# Natural Language Embedded Programs for Hybrid Language Symbolic Reasoning

## Abstract

How can we perform computations over natural language representations to solve tasks that require symbolic and numeric reasoning? We propose *natural language embedded programs* (NLEP) as a unifying framework for addressing math/symbolic reasoning, natural language understanding, and instruction following tasks. Our approach prompts a language model to generate full Python programs that define functions over data structures which contain natural language representations of structured knowledge. A Python interpreter then executes the generated code and prints the output. Despite using a task-general prompt, we find that this approach can improve upon strong baselines across a range of different tasks including math and symbolic reasoning, text classification, question answering, and instruction following. We further find the generated programs are often interpretable and enable post-hoc verification of the intermediate reasoning steps.

**Instruction:** Who are the US presidents elected after 1950 and born on Monday?

```python
# Step 1: Import necessary libraries                          NLEP generated by GPT-4
from datetime import datetime

# Step 2: Define a dictionary storing the knowledge about US presidents
us_presidents = {
    "Dwight D. Eisenhower": {"birth_date": "1890-10-14", "term_start": "1953-01-20"},
    "John F. Kennedy":      {"birth_date": "1917-05-29", "term_start": "1961-01-20"},
    "Lyndon B. Johnson":    {"birth_date": "1908-08-27", "term_start": "1963-11-22"},
    "Richard Nixon":        {"birth_date": "1913-01-09", "term_start": "1969-01-20"},
    "Gerald Ford":          {"birth_date": "1913-07-14", "term_start": "1974-08-09"},
    "Jimmy Carter":         {"birth_date": "1924-10-01", "term_start": "1977-01-20"},
    "Ronald Reagan":        {"birth_date": "1911-02-06", "term_start": "1981-01-20"},
    "George H. W. Bush":    {"birth_date": "1924-06-12", "term_start": "1989-01-20"},
    "Bill Clinton":         {"birth_date": "1946-08-19", "term_start": "1993-01-20"},
    "George W. Bush":       {"birth_date": "1946-07-06", "term_start": "2001-01-20"},
    "Barack Obama":         {"birth_date": "1961-08-04", "term_start": "2009-01-20"},
    "Donald Trump":         {"birth_date": "1946-06-14", "term_start": "2017-01-20"},
    "Joe Biden":            {"birth_date": "1942-11-20", "term_start": "2021-01-20"},
}

# Step 3: Define a function that finds the presidents who are elected after a certain year and born on a certain day
# of the week
def find_presidents(presidents, year, day_of_week):
    result = []
    for president, info in presidents.items():
        birth_date = datetime.strptime(info["birth_date"], "%Y-%m-%d")
        term_start = datetime.strptime(info["term_start"], "%Y-%m-%d")
        if birth_date.strftime("%A") == day_of_week and term_start.year > year:
            result.append(president)
    return result

# Step 4: Print an answer in natural language using the knowledge and function defined above
year = 1950
day_of_week = "Monday"
presidents = find_presidents(us_presidents, year, day_of_week)
if presidents:
    print("The US presidents who were elected after 1950 and born on Mondays are: " + ", ".join(presidents) + ".")
else:
    print("There are no US presidents who were elected after 1950 and born on Mondays.")
```

The left margin labels the code blocks: Tool Using (Step 1), Structured knowledge (Step 2), Function definition (Step 3), Print answer (Step 4).

**Stdout:** The US presidents who were elected after 1950 and born on Mondays are: Gerald Ford, Ronald Reagan, Bill Clinton.

**Figure 1:** A generated NLEP correctly answers the given question while ChatGPT-4 obtains an incorrect answer (link). This NLEP uses the date-weekday conversion tool in the `datetime` package, constructs structured knowledge about US presidents, implements a selection function, and outputs natural language responses depending on the function output. A more detailed comparison between NLEP and ChatGPT-4 code interpreter is shown in Figure 5.

# 1 INTRODUCTION

Solving complex language tasks often requires performing computations over natural language representations. For language-based reasoning, chain-of-thought prompting (CoT; Wei et al., 2022) has emerged as a promising approach for surfacing the symbolic reasoning capabilities of large language models (LLMs). However, certain types of computations (e.g., arithmetic) are unnatural to perform in pure language space, and hence present difficulties for LLMs. General-purpose programming languages, on the other hand, provide convenient abstractions as well as predefined libraries and functions for natively implementing many types of symbolic computations, and there has been much recent work on interleaving program calls within CoT-style reasoning to extend the capabilities of LLMs. While promising, existing methods are generally limited to narrow types of tasks such as math and symbolic reasoning (Chen et al., 2022; Cai et al., 2023; Gao et al., 2023), simple API calling (Schick et al., 2023; Paranjape et al., 2023; Liang et al., 2023a), and database accessing (Cheng et al., 2022). These works moreover rely on task-specific prompts which are hard to generalize across datasets.

This work describes a task-general approach for combining the language-based reasoning capabilities of LLMs with symbolic computations enabled by the use of programs. Specifically, we prompt LLMs to generate *natural language embedded programs* (NLEPs), which are fully executable Python programs containing appropriate package importing, structured natural language representations of knowledge, function definitions for problem solving, and response printing. The generated NLEP is then executed using a Python interpreter that captures the standard output of the program as the response. An example of an NLEP generated by GPT-4 is shown in Figure 1.

NLEPs use code as a scaffold to reason over natural language representations of knowledge. This makes our approach different from ToolFormer (Schick et al., 2023) and language model as tool maker (LATM; Cai et al., 2023), which instead use language as the scaffold and interleave API calls within natural language sentences during LLM generation. Compared to program-of-thought (PoT; Chen et al., 2022) and program aided language models (PAL; Gao et al., 2023), which mainly focus on math and symbolic problems, NLEPs utilize more flexible programming elements including packages, data types/structures, and functions. This design allows NLEP to solve more general tasks such as question answering over factual knowledge. Existing works also generally require dataset-specific prompts (e.g., to demonstrate tool usage). In contrast, we find that we can generate NLEPs for various tasks by feeding task-general demonstrations (e.g., the same demonstrations for all classification tasks) as prompts to an LLM.

Experiments across math and symbolic reasoning, question answering and instruction following, and text classification tasks demonstrate that NLEPs can potentially serve as a unifying framework for tackling a variety of tasks within the prompt-based learning framework. In particular, our results suggest that appropriately-prompted LLMs can make rich use of programming structures to tackle tasks that require a combination of language-based reasoning and symbolic computations.

# 2 APPROACH: NLEP PROMPTING

In this section we describe *natural language embedded programs* (NLEPs) in more detail and present a simple prompting framework for NLEP generation. We also describe instantiations of NLEPs for different types of tasks.

**Natural language embedded programs (NLEPs).** An NLEP is a program containing both programming code and natural language. NLEPs use natural language in several different ways. First, it uses natural language comments to guide step-by-step program generation. Second, language is used to represent structured knowledge through Python's native data structures (e.g., dictionaries and lists). Finally, an NLEP uses language to print fluent responses to the user input by constructing a standard output string containing references to program variables.

The hybrid language-symbolic design of NLEP enables generalized problem solving for natural language, math, symbolic reasoning, and API calling tasks, which have traditionally been tackled by separate mechanisms. This approach combines the benefits of language-based reasoning with program synthesis: comments and knowledge in natural language improve program generation, while the structured/symbolic reasoning powered by program interpreters provides more accurate computations than would have been obtained via direct decoding from LLMs.

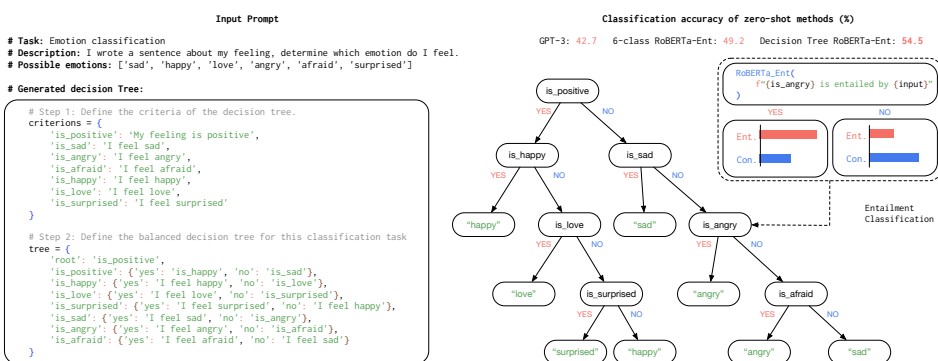

**Figure 2:** A decision tree structure generated within an NLEP for emotion classification based on task description using an example program for SST2 as the prompt. The branching of each node is decided by a RoBERTa (Liu et al., 2019) text entailment model. This language-based decision tree generated by an NLEP outperforms GPT-3 and entailment-based multi-class prediction (Ge et al., 2023) without needing any task-specific examples (i.e., exemplars specific to the emotion classification dataset).

An example of an NLEP for answering a question is shown in Figure 5. In the generated program, each section is preceded by comments in natural language, and the defined counting function uses knowledge stored in a key-value dictionary (which itself is generated from GPT-4's internal knowledge) to find the correct answer. Finally, the answer is printed through a natural language response. In this example, we generated 5 independent NLEPs and found that they achieve 100% accuracy, compared to 60% for ChatGPT-4 and 40% GPT-4 API.

**NLEP structure.** More generally, each NLEP contains four sections: (1) importing necessary libraries, (2) defining variables containing structured knowledge, (3) implementing problem-solving functions, and (4) printing the response in natural language. Instead of providing direct solutions for each task, we guide the model to arrive at a solution following this four-step process. As show in the example in Figure 1, an NLEP answers the question by constructing a structured knowledge dictionary containing the birthday and start date of the US presidents. To recognize the weekdays, the program utilizes pre-defined functions in the `datetime` package. The selected answers are stored in a `list` and then embedded into an output template. The NLEP also handles the situation when no answer is found. The correct answer is then printed by the NLEP.

**Task-general demonstration prompts.** As is standard in chain-of-thought prompting (Nye et al., 2021; Wei et al., 2022), our approach uses demonstration prompts for NLEP generation. However, unlike previous approaches our demonstrations are not task-specific. For example, for all classification tasks we consider we use the *same* demonstration prompt (derived from SST2). Similarly, we use mostly the same prompt for our math and symbolic reasoning tasks. This task-general prompt is similar in spirit to zero-shot chain-of-thought prompting (Kojima et al., 2023) which adds a task-agnostic prompt ("`Let's think step-by-step`") to elicit the reasoning capabilities of LLMs in a task-agnostic way. The prompts used for the various tasks are given in Table 1, and the exact prompts are given in Appendix C. In summary, we use 4 different demonstration prompts across 16 tasks, each of which works well within a task category. Thus, while the proposed method is not fully task-agnostic in the strictest sense of the term, it is still more flexible than previous approaches that combine program synthesis with chain-of-thought prompting (Chen et al., 2022; Gao et al., 2023), which use examples from the dataset to craft prompts.

**Programmatic reasoning for natural language understanding tasks.** Prior works on combining program synthesis with LLM-based reasoning have generally focused on math and symbolic reasoning tasks (Chen et al., 2022; Gao et al., 2023), and it has not been clear how such methods could be extended to address natural language understanding (NLU) tasks. We show that NLEPs can be straightforwardly extended to tackle more language-based tasks.

For question answering, we apply NLEP prompting and the target output is constructed by the generated programs. Classification tasks, on the other hand, are handled by a different type of NLEP consisting of a decision tree. Each node of the decision tree is annotated by a simple natural language sentence, and the Yes/No decisions at each node are handled in a zero-shot way by an

| Domain | Task | Split | Dataset Size | Prompt | Output format |
|--------|------|-------|--------------|--------|---------------|
| Math and Symbolic Reasoning | Tracking Shuffled Objects (7) | test | 250 | C.1 | Option |
| | Dyck Language | test | 250 | C.1 | Free Form |
| | Word Sorting | test | 250 | C.1 | Free Form |
| | Chinese Remainder Theorem | test | 250 | C.1 | Number |
| | Scheduling Meeting | test | 250 | C.1 | Free Form |
| | GSM-Hard | test | 1319 | C.1 | Number |
| | Game of 24 | test | 100 | C.2 | Free Form |
| Question Answering | StrategyQA | dev | 229 | C.1 | Yes/No |
| | TruthfulQA | test | 817 | C.3 | Free Form |
| | VicunaQA | test | 80 | C.3 | Free Form |
| Text Classification | SST2 | val | 872 | C.4 | Class |
| | Cola | val | 1.04k | C.4 | Class |
| | Emotion-Classification | val | 2k | C.4 | Class |
| | Amazon Review | val | 5k | C.4 | Class |
| | Hate-Speech | val | 478 | C.4 | Class |
| | Social Bias Frame | val | 16.7k | C.4 | Class |

**Table 1:** Summary descriptions of the various tasks considered in this work.

entailment classifier, which has in general been shown to be an effective approach to zero-shot text classification (Obamuyide & Vlachos, 2018; Condoravdi et al., 2003; Ge et al., 2023). Concretely, given the tree we compute the entailment score between the input and the language description of each node and traverse the decision tree until a leaf node is reached. We emphasize that the topology of the tree and the language description of each node is generated by the prompted LLM. The demonstration prompt for classification tasks is given by a manually constructed example for SST2 (Wang et al., 2018). We find that this prompt can generate NLEPs containing sensible decision trees for various classification tasks without requiring task-specific examples. An example of the generated program and the corresponding decision tree is shown in Figure 2.

## 3 EXPERIMENTS

We evaluate natural language embedded programs (NLEPs) on 16 tasks across three broad task categories. The tasks and corresponding prompts are summarized in Table 1.

**Math and symbolic reasoning** tasks include Tracking Shuffled Objects (7), Dyck Language, Word Sorting and Chinese Remainder Theorem from BigBench (Srivastava et al., 2023), Scheduling Meeting task from Cai et al. (2023), GSM-Hard benchmark of math word problems from Gao et al. (2023), and Game of 24 (Yao et al., 2023a). We use two examples for all tasks except for Game of 24, for which we applied a word sorting example to elicit stronger game-playing reasoning ability. The exact NLEP prompts we used are given in Appendix C.1 and C.2.

**Question answering** tasks include the StrategyQA (Geva et al., 2021a), TruthfulQA (Lin et al., 2022), and VicunaQA (Chiang et al., 2023) benchmarks. StrategyQA requires models to answer multi-hop questions with "Yes" or "No". TruthfulQA and VicunaQA contain questions and instructions requiring free-form responses. VicunaQA also allows us to test how NLEPs perform in the popular instruction-following setting. The evaluation metrics on question answering focus on accuracy, relevance, and factuality of the generated answers. The prompts in Appendix C.1 are used for StrategyQA. For TruthfulQA and VicunaQA, we added an example with a longer response to encourage more detailed response generation (Appendix C.3).

**Text classification** tasks includes tasks that require understanding of both natural language inputs and labels. We evaluate NLEP on movie-review classification (SST2; Socher et al., 2013), linguistic acceptance (COLA; Warstadt et al., 2019), emotion classification (Saravia et al., 2018), amazon review (Ni et al., 2019), hate speech detection (de Gibert et al., 2018), and stereotypes recognition (Sap et al., 2019). We use the prompts in Appendix C.1 for model-free classification. For decision tree generation, the prompts in Appendix C.4 are applied.

### 3.1 MATH AND SYMBOLIC REASONING

We compare NLEP prompting with chain-of-thought (CoT; Wei et al., 2022), program-of-thought (PoT; Chen et al., 2022), and LLMs as tool makers (LATM; Cai et al., 2023). We also compare

against tree-of-thought (ToT; Yao et al., 2023a) on the Game of 24 benchmark, where ToT outperforms CoT by a significant margin (but requires many more calls to the LLM). We evaluate CoT and PoT with both task-general and task-specific demonstrations. Since LATM needs in-domain input-output pairs to create tools, we only report the results with task-specific examples for LATM.

**Task-general prompting.** For task-general prompts we use two examples as the in-context demonstration for the math and symbolic reasoning benchmarks (see Table 1 and Appendix C). For CoT, we present two examples with intermediate reasoning represented in natural language rather than as programs. Our task-general PoT implementation takes the math and symbolic reasoning lines similar as Chen et al. (2022) and Gao et al. (2023), but without the step-by-step programming scheme in NLEP as an ablation.

**Task-specific prompting baselines.** We report the task-specific prompting performance as an "upper bound" for each task. For CoT, we use the same prompting settings (from 3 to 8-shot) adopted in previous studies (Cobbe et al., 2021; Cai et al., 2023; Fu et al., 2023). For PoT, we use the same in-context examples as in the task-specific CoT examples, but provide intermediate reasoning steps in Python code. On the GSM-Hard benchmark, we adopt the demonstrations (9-shot) for GSM8K used in Chen et al. (2022). For the Chinese Remainder Theorem and Scheduling Meeting benchmarks, we construct the in-context examples with the first three successful instances of task-general PoT. For LATM, we evaluate its performance on Tracking Shuffled Objects (7) using the provided tool and cite the results for other tasks from Cai et al. (2023). Details are shown in Appendix D.

Program synthesis approaches (PoT and NLEP) may sometimes generate non-executable programs if lack task-specific programming demonstration. For both approaches, we select certain benchmarks to resample up to three additional programs if the returned program failed at execution. Since this condition is triggered only if program execution fails, there is no label leakage. We discuss this further in Section 4 and provide results details in Appendix B.

| Tasks / Method | GPT-4 | | | | | | GPT-3.5-Turbo | | | | |
| --- | --- | --- | --- | --- | --- | --- | --- | --- | --- | --- | --- |
| | (a) Task-Specific | | | (b) Task-General | | | (c) Task-Specific | | (d) Task-General | | |
| | CoT | PoT | LATM | CoT | PoT | NLEP | CoT | PoT | CoT | PoT | NLEP |
| Tracking Shuffled Objects | **100.0** | **100.0** | **100.0** | 81.2 | 98.4 | **100.0** | 68.0 | 6.8 | 51.2 | 88.4 | 74.4 |
| Dyck Language | 63.6† | 60.8 | 87.5† | 39.6 | 66.4 | **91.6** | 20.4† | 28.4 | 38.0 | 4.0 | 7.2 |
| Word Sorting | 90.9† | **100.0** | 99.1† | 84.4 | 99.6 | 99.6 | 59.2† | **100.0** | 75.2 | **100.0** | 99.6 |
| Chinese Remainder Theorem | 0.0† | **100.0** | **100.0**† | 0.0 | 84.4 | 97.2 | 0.0† | **100.0** | 0.0 | 72.4 | 96.4 |
| Scheduling Meeting | 55.6† | 75.2 | **100.0**† | 82.8 | 85.2 | 93.2 | 18.9† | 33.6 | 39.6 | 49.2 | 85.6 |
| GSM-Hard | 57.4 | **74.1** | – | 54.9 | 69.3 | 67.7 | 45.0 | 63.4 | 42.8 | 52.2 | 54.1 |
| **Average** | 61.3 | 85.0 | **97.3** | 57.2 | 83.9 | 91.6 | 35.3 | 55.4 | 41.1 | 61.0 | 69.6 |

**Table 2:** Performance on math and symbolic reasoning tasks with both task-specific and task-general demonstration prompts. † stands for results from Cai et al. (2023). LATM results are not available for GSM-Hard benchmark as it is hard to derive a generally applicable tool function for all test cases.

### 3.1.1 RESULTS

We show the main results of NLEP prompting on six math and symbolic reasoning tasks in Table 2. An example of NLEP generated for solving a Dyck language problem is shown in Figure 3(a).

**GPT-4 Results.** Among the three approaches which use task-general prompts, NLEP outperforms both CoT and PoT on 5 of 6 tasks. The large performance gap between NLEP and CoT suggests that programmatic reasoning can enable more accurate answers. Compared to PoT, NLEP achieves significantly higher average accuracy, especially on the Dyck Language (66.4%→91.6%) and the Chinese Remainder Theorem (84.4%→97.2%) tasks. On GSM-Hard, we confirm the same phenomenon discovered by Gao et al. (2023) where language does not further benefit the calculation accuracy with GPT-4.

NLEP also achieves comparable performance to task-specific, few-shot prompting methods. Notably, our method achieves the best performance on Tracking Shuffled Objects (7) and Dyck Language, and outperforms task-specific CoT on many benchmarks. On the Word Sorting benchmark, NLEP only fails on one instance where the input word sequence contains "steelmake" and GPT-4 automatically corrected it to "steelmaker". We find that the high scores of task-specific PoT on

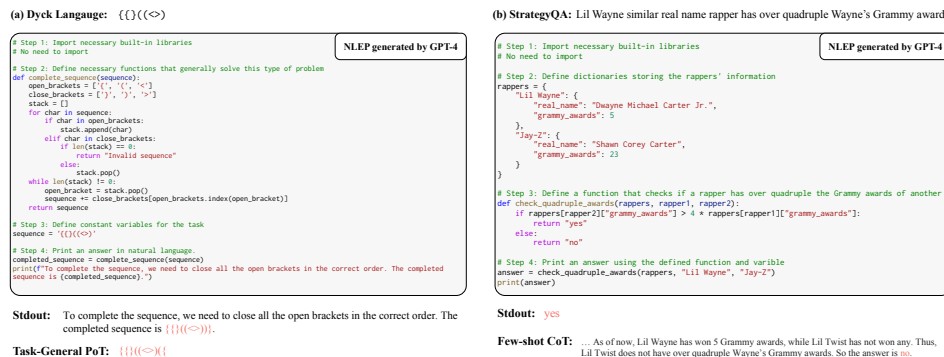

**Figure 3:** NLEP generated for solving Dyck language and StrategyQA problems. For Dyck, the instruction is *"Complete the rest of the sequence, making sure that the parentheses are closed properly."* For StrategyQA, the instruction is *"Answer the question with yes or no."*

Word Sorting and Chinese Remainder Theorem come from the generally applicable programming code from the in-context demonstrations.

**GPT-3.5 Results.** We observe significant performance degradation with GPT-3.5, presumably due to its limited programming capabilties. However NLEP still achieves the best average performance, exhibiting significant improvement on 5 of 6 tasks over all baselines. On the Dyck Language benchmark, program-based strategies (PoT and NLEP with task-general prompts) failed to accomplish the problem without task-specific examples, highlighting the need for strong backbone LLMs.

| setting | | | Task-Specific | | | | Task-General | |
|---|---|---|---|---|---|---|---|---|
| | IO | CoT | IO (best of 100) | CoT (best of 100) | ToT (b=1) | ToT (b=5) | PoT | NLEP (ours) |
| Game of 24 (%) | 7[†] | 4[†] | 33[†] | 49[†] | 45[†] | **74**[†] | 52 | 66 |

**Table 3:** Performance on the Game of 24 benchmark. CoT and ToT stand for chain-of-thought (Wei et al., 2022) and tree-of-thought (Yao et al., 2023a) prompting respectively. [†] shows the results from Yao et al. (2023a).

**Game of 24 results.** Table 3 shows the results on the challenging Game of 24 task from Yao et al. (2023a). Our approach also surpasses the oracle setup of IO/CoT, which calculates the success rate of IO/CoT by considering the best of 100 samples for each instance. However, unlike ToT which requires in-context demonstrations for each decomposed sub-task, NLEP prompting achieves a significant performance gain over ToT (b=1) without requiring a computationally expensive multi-chain inference procedure.

## 3.2 Question Answering and Instruction Following

We next apply NLEP prompting to tackle question answering and instruction following tasks requiring different answer forms: StrategyQA, TruthfulQA, and VicunaQA. StrategyQA tests for commonsense reasoning ability of language models and requires Yes/No answers, while TruthfulQA and VicunaQA have free-form responses.

| setting | GPT-4 | | | | GPT-3.5-Turbo | | | |
|---|---|---|---|---|---|---|---|---|
| | Task-specific | Task-general | | | Task-specific | Task-general | | |
| | CoT | CoT | PoT | NLEP (ours) | CoT | CoT | PoT | NLEP (ours) |
| StrategyQA | **81.7** | 78.6 | 68.6 | 81.2 | 71.6 | 68.1 | 50.2 | 61.1 |

**Table 4:** Performance on the StrategyQA benchmark. The experimental setup is the same as in Table 2. Note that LLMs do not always generate "Yes" or "No". and we only predict the "Yes" label if the "Yes" string is generated explicitly. See Appendices C.1 and D for implementation details.

**StrategyQA.** Experiment results are presented in Table 4. With GPT-4, NLEP achieves the best performance under the task-general prompt setting and is competitive with the task-specific CoT.

With GPT-3.5, although the scores of code-based strategies decrease more than CoT (PoT: 18.4%, NLEP: 20.1%, task-general CoT: 10.5%, task-specific CoT: 10.1%), NLEP still exceeds PoT by a significant margin. An example of output is shown in 3(b).

**TruthfulQA.** We also evaluate how NLEP prompting influences the factuality of question answering with the TruthfulQA benchmark (Lin et al., 2022). A fine-tuned GPT-3 model is applied for automatic scoring. In this experiment, we compare the vanilla auto-regressive text generation method against NLEP. Traditionally, such question answering tasks have been solved only with black-box language model without explicit symbolic computations due to the complexity of test questions.

| Foundation Model | Mode | True | Info | True * Info |
|---|---|---|---|---|
| GPT-4 | Text | **76.01** | 97.55 | 73.56 |
| | NLEP | 75.76 | **99.63** | **75.40** |
| GPT-3.5-Turbo | Text | 68.91 | 98.90 | 67.93 |
| | NLEP | 61.69 | 97.18 | 59.00 |

**Table 5:** Performance of GPT-4 and GPT-3.5-Turbo on the TruthfulQA benchmark.

The results are shown in Table 5. With GPT-4, the truth score of NLEP prompting strategy is close to standard LLM-based generation, while the informativeness score is higher. However, performance degrades significant with GPT-3.5-Turbo, indicating a strong dependence on the programming ability of the underlying language model.

| Model | # NLEP >Text | Detail | % Score | % Length Bias |
|---|---|---|---|---|
| GPT-4 | 23.75 | yes | 93.08 | 72.72 |
| | | no | **105.06** | 26.67 |
| GPT-3.5 -Turbo | 38.75 | yes | 101.22 | **3.13** |
| | | no | 102.50 | 10.34 |

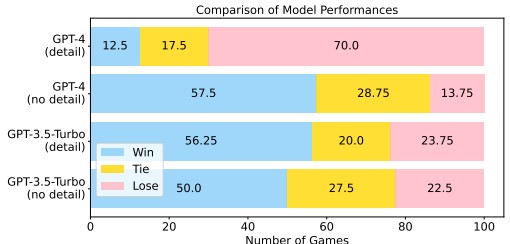

**Figure 4:** Automatic evaluation results of NLEP against standard LLM-based generation with different models. **# NLEP >Text** means that the % of NLEP responses containing more tokens than the baseline. **Detail** means if the evaluation metric considers details and response lengths. **Score** stands for the scores received by NLEP divided by the baseline scores (>100 means NLEP is better). **Win**, **tie**, and **lose** stand for the % of evaluation cases resulting in each category. **Length Bias** shows how much the evaluation pipeline prefers longer or shorter answers (lower means fairer, introduced in Appendix E.3).

**VicunaQA.** Results on the VicunaQA benchmark are shown in Figure 4, where we follow the standard approach and evaluate the answers using GPT-4. We find that GPT-4 prefers its own generations, which are generally more detailed than GPT-3.5-Turbo and NLEP responses. To control for the bias due to response lengths, we also assess all responses without the requirement about details using another evaluation prompt. The evaluation prompts with and without the requirement on details is shown in Appendix E.1 and E.2.

As we demonstrate in Figure 4, this assessment leads to different results on GPT-4. After removing the detail requirement in the automatic scoring pipeline, NLEP achieves better performance. This suggests that NLEP can help GPT-4 generate accurate, factual, and relevant responses. However, human-generated programs for pretraining the GPT-4 models usually do not embed long pieces of natural language. As a result, the responses generated by NLEP have a limited level of detail. We also notice that NLEP improves GPT-3.5-Turbo under both detail assessment settings, since neither text and NLEP generated by GPT-3.5-Turbo reaches the detail level preferred by the GPT-4 scorer.

## 3.3 TEXT CLASSIFICATION

Finally, we evaluate whether NLEPs can be applied to solve text classification tasks that have traditionally been difficult for pure program synthesis-based approaches. As discussed in section 2, we manually construct a decision tree NLEP for SST2 and use it as a prompt to guide GPT models to generate decision trees for other tasks only with task and label descriptions. An example input and output NLEP generated by GPT-4 for emotion classification is shown in Figure 2.

| Model | Method | Performance (Num. Classes) | | | | | |
|-------|--------|----------|------------|-----------|---------|----------|---------|
| | | cola (2) | emotion (6) | amazon (5) | hsd (2) | sbic (3) | **Average** |
| RoBERTa | Multi-class Prompting | 65.87 | 49.2 | 33.31 | 67.78 | 52.99 | 53.83 |
| | Human-Generated Tree | **69.03** | 22.20 | 26.88 | 64.85 | **58.37** | 48.27 |
| | NLEP w/ GPT-3.5 | 56.66 | 35.1 | 33.46 | 67.36 | 38.25 | 46.17 |
| | NLEP w/ GPT-4 | 68.94 | **54.5** | **38.88** | **70.92** | 55.95 | **57.65** |
| DeBERTa | Multi-class Prompting | 53.50 | 51.93 | 37.01 | 67.78 | 59.08 | 53.86 |
| | Human-Generated Tree | **69.22** | 32.15 | 33.00 | **72.18** | 55.02 | 52.31 |
| | NLEP w/ GPT-3.5 | 49.66 | 39.00 | 36.18 | 70.29 | 52.49 | 49.52 |
| | NLEP w/ GPT-4 | 68.36 | **55.4** | **40.2** | 70.08 | **59.68** | **58.74** |
| None | Model-free NLEP w/o Tree | 69.13 | 40.55 | 25.76 | 59.62 | 37.63 | 46.54 |

**Table 6:** Zero-shot performance of different human-crafted and LLM-generated text classification schemes. The GPT-4 generated decision trees consistently exhibit significant improvement. For model-free NLEP, generated code can be executed on the entire validation set in 2 seconds and notably surpasses the random baseline, with cola notably matching the state-of-the-art performance.

| | Model-free | RoBERTa-Manual | RoBERTa-Automatic | DeBERTa-Manual | DeBERTa-Automatic |
|------|-----------|----------------|-------------------|----------------|-------------------|
| SST2 | 66.17 | 83.03 | 87.36 | 84.06 | 83.49 |

**Table 7:** Performance of manually crafted vs. generated decision trees on SST2.

We compare NLEP against two baseline methods. Our first baseline uses the zero-shot classification method proposed in Ge et al. (2023) ("multi-class prompting"). This method uses the same entailment models but makes the prediction without the tree structure. Our second baseline asks a human expert to design a decision tree for each task also based on the SST-2 example. The results shown in Table 6 show that NLEP generated by GPT-4 outperforms multi-class prompting and human-generated tree baselines on most datasets.

Since we use the trees derived from SST2 to prompt the LLM for the classification tasks, it would be inappropriate to use these examples for SST2 itself. For SST2, we thus use an automatically generated decision tree for the CoLA task to prompt GPT-4 to generate a new tree for SST2. As shown in Table 7, the automatically generated tree matches the performance of the SST2 decision tree created by the authors.

**Model-free NLEP**. We also tried using the task-general prompt shown in C.1 to generate NLEPs that directly use programs to solve these tasks. These programs do not need any neural models and are hence very efficient (e.g., finishing the entire validation set in about 2 seconds on CPUs). The results can be found in Table 6 ("Model-free NLEP"). While not achieving the performance of entailment-based methods, the generated NLEP significantly outperforms random baselines, suggesting that this may be a viable approach for quickly extracting simple and interpretable classifiers from LLMs.

## 4 DISCUSSION

**Execution failures and retries.** While the task-general PoT and NLEP lack programming demonstrations for the target task, GPT-4 in general is able to generate bug-free programs as presented in Appendix B Table 11. Notably, both PoT and NLEP obtain execution error rate of 0 on Tracking Shuffled Objects (7) and Word Sort tasks. One advantage of the program synthesis approaches such as PoT and NLEP is that non-executable programs can be identified and filtered. This gives LLMs the chance to "self-correct" and generate new answers, and we take advantage of this in our math and symbolic reasoning tasks by generating up to three programs if there is an execution failure on certain benchmarks. (For fair comparison we apply this reattempting scheme to PoT as well). We ablate on this mechanism in Appendix B, Tables 8, 10 and 11. Besides effectively reducing the execution error as presented in Table 11, these retries greatly enhance the reasoning accuracy. In particular, 12% and 15.6% improvement is observed on the Chinese Remainder Theorem and the Scheduling Meeting tasks in Table 8(b). In this work we only experiment extra retries with larger temperatures for diverse sampling and leave more advanced "self-correction" algorithms (e.g., those that make use of error messages (Cai et al., 2023; Hu et al., 2023)) for future work.

**Different foundation LLMs for NLEP.** The large performance gaps of CodeLlama-7b-Instruct (Rozière et al., 2023), GPT-3.5-Turbo, and GPT-4 in Table 2 and 9 indicate that strong programming ability of underlying LLMs is vital to generate accurate responses and achieve performance improvements with NLEP. For example, on the Dyck Language task, GPT-3.5-Turbo only achieves 7.2% accuracy while GPT-4 achieves 91.6% accuracy. TruthfulQA experiments also show that NLEP could *hurt* the factuality of GPT-3.5-Turbo. Surprisingly, zero-shot CodeLlama-7b (Rozière et al., 2023) trained using NLEP-style data (without in-domain examples) demonstrates superiority on Tracking Shuffled Objects (7) benchmark over NLEP prompted GPT-3.5 and Word Sorting benchmark over task-general CoT prompted GPT-3.5, even with significantly fewer parameters (see details in Appendix B). It shows the potential for effective training of compact large language models, enabling them to achieve performance levels comparable to those of extremely large models.

**Limitation of NLEP prompting.** We found that the NLEP prompts are not suitable for generating long-form natural language responses. Experimental results on VicunaQA show that most responses generated by NLEP prompting have fewer tokens than responses obtained from usual LLM generation. This feature is expected, because most naturally-occurring programs (on which the LLMs were pretrained) do not contain large chunks of natural language. Future work could consider incorporating (potentially synthetically generated) programs with longer-form natural language within the pretraining set to enable the application of NLEP to more involved NLG tasks.

## 5    RELATED WORK

**Large language models for reasoning.** State-of-the-art LLMs (OpenAI, 2022; 2023; Touvron et al., 2023; Zeng et al., 2022) have shown very strong performance on complicated reasoning tasks, including commonsense (Geva et al., 2021b), math (Cobbe et al., 2021), symbolic reasoning (Suzgun et al., 2022), and programming (Austin et al., 2021; Chen et al., 2021). Tackling such tasks with LLMs often requires prompting them with demonstrations that elicit their reasoning capabilities. Wei et al. (2022) proposed chain-of-thought prompting technique that encourages language to generate answers step-by-step. Wang et al. (2022) found that self-consistency can further improve the performance of chain of thoughts reasoning ability. Kojima et al. (2023) discovered that LLMs can perform reasoning without any demonstrations through adding the incantation "`Let's think step-by-step`". Tree of thoughts (Yao et al., 2023a) and graph of thoughts (Yao et al., 2023b; Besta et al., 2023) were proposed to tackle tasks that require more complicated reasoning processes. These improved reasoning methods apply chain of thoughts as the atomic reasoning step but organize reasoning "chains" through more advanced mechanisms.

**Programs and tools.** Previous studies have found that some limitations of LLMs can be overcome by combining program synthesis techniques with prompt-based learning. Program of thoughts (Chen et al., 2022) and program aided language model Gao et al. (2023) both translate mathematical questions to equations and use the python interpreter to ensure the correctness of the calculations. Another line of related work for enabling LLMs to use tools is through interleaving API calls during LLM generation (Qin et al., 2023; Liang et al., 2023b; Mialon et al., 2023; Tang et al., 2023). APIs can aid many tasks that are challenging for LLMs by providing tailored tools (e.g., calculators, search) that can solve specific tasks. Toolformer (Schick et al., 2023) addresses reasoning tasks by using predefined tools, and LLMs as tool makers (LATM) can implement functions solving a class of tasks based on few-shot examples (Cai et al., 2023). With these solutions, the correctness of a prediction can be ensured if correct API is called and correct inputs are selected. Existing works on combining program synthesis and tool usage with LLMs generally rely on task-specific prompts, in contrast to the more task-general prompt explored in the present work.

## 6    CONCLUSION

This work describes natural language embedded programs (NLEP), which flexibly combine natural language reasoning with program synthesis within prompt-based learning to tackle a variety of tasks. Our experiments demonstrate that NLEPs expand the scope of applications that can be addressed by program synthesis by more closely incorporating natural language during code generation.

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

# A   ADDITIONAL EXAMPLE COMPARING WITH CHATGPT-4

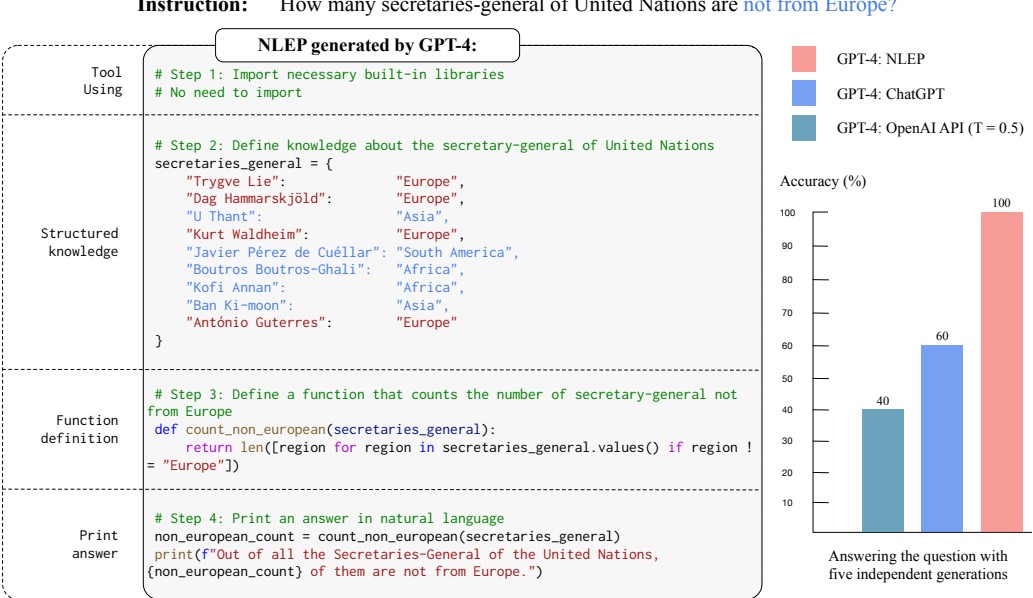

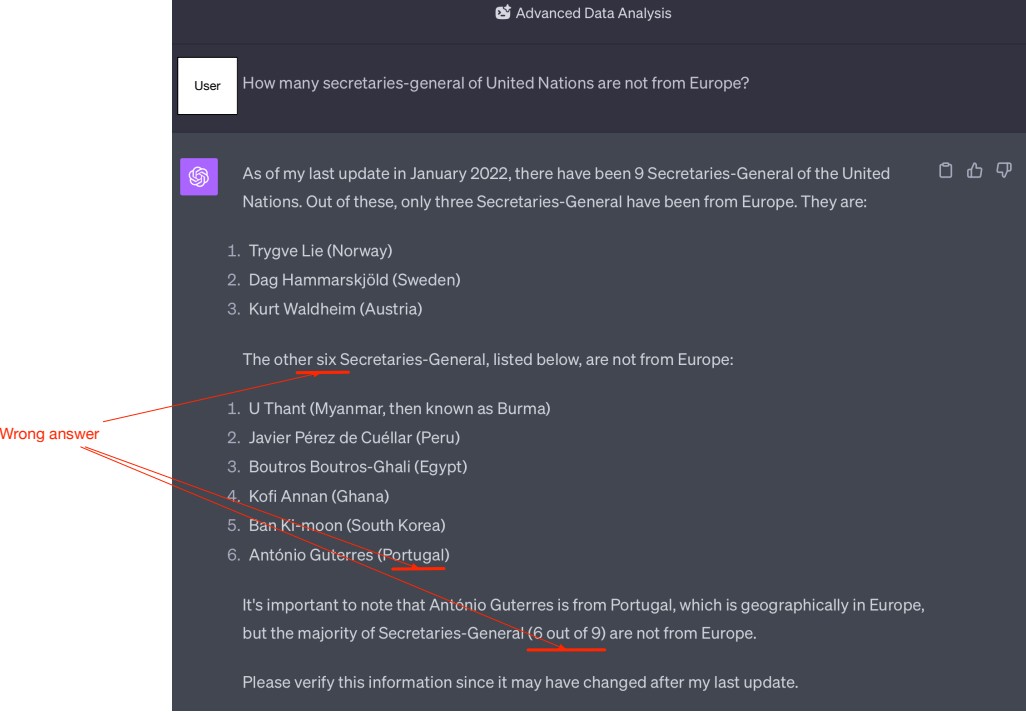

**Figure 5:** NLEP answering a question which requires numeric reasoning of structured knowledge. ChatGPT-4 code interpreter (currently the advanced data analysis option) constantly prefers to answer this question with plain natural language.

## B  ADDITIONAL RESULTS AND ANALYSIS ON MATH AND SYMBOLIC REASONING

We present the detailed experimental results of math and symbolic reasoning tasks in Tables 8 to 10, with execution failure analysis in Table 11.

**GPT Results**. We report the results of Table 2 with more details in Table 8. The effect of extra retries described in Section 4 is highlighted with ($\rightarrow$).

| | Tracking Shuffled Objects (7) | Dyck Language | Word Sorting | Chinese Remainder Theorem | Scheduling Meeting | GSM-Hard |
|---|---|---|---|---|---|---|
| (a) Task-Specific Prompting: GPT-4 | | | | | | |
| CoT | **100.0** | 63.6$^\dagger$ | 90.9$^\dagger$ | 0.0$^\dagger$ | 55.6$^\dagger$ | 57.4 |
| PoT | **100.0** | 60.8 | **100.0** | **100.0** | 75.2 | **74.1** |
| LATM | **100.0** | 87.5$^\dagger$ | 99.1$^\dagger$ | **100.0**$^\dagger$ | **100.0**$^\dagger$ | - |
| (b) Task-General Prompting: GPT-4 | | | | | | |
| CoT | 81.2 | 39.6 | 84.4 | 0.0 | 82.8 | 54.9 |
| PoT | 98.4 | 66.4 | 99.6 | 76.4 ($\rightarrow$84.4) | 84.4 ($\rightarrow$85.2) | 69.3 |
| NLEP (Ours) | **100.0** | **91.6** | 99.6 | 85.2 ($\rightarrow$97.2) | 77.6 ($\rightarrow$93.2) | 67.7 |
| (c) Task-Specific Prompting: GPT-3.5-Turbo | | | | | | |
| CoT | 68.0 | 20.4$^\dagger$ | 59.2$^\dagger$ | 0.0$^\dagger$ | 18.9$^\dagger$ | 45.0 |
| PoT | 6.8 | 28.4 | **100.0** | **100.0** | 33.6 | 63.4 |
| (d) Task-General Prompting: GPT-3.5-Turbo | | | | | | |
| CoT | 51.2 | 38.0 | 75.2 | 0.0 | 39.6 | 42.8 |
| PoT | 88.4 | 4.0 | **100.0** | 58.4 ($\rightarrow$72.4) | 46.8 ($\rightarrow$49.2) | 39.0 ($\rightarrow$52.2) |
| NLEP (Ours) | 74.4 | 7.2 | 99.6 | 94.8 ($\rightarrow$96.4) | 75.2 ($\rightarrow$85.6) | 50.9 ($\rightarrow$54.1) |

**Table 8:** Performance on six reasoning tasks. $^\dagger$ stands for results from LATM (Cai et al., 2023). Results with $^\dagger$ and of LATM are reported on the last 240 instances with the first 10 instances as training and validation sets for tool making according to LATM's design. LATM is not appropriate for GSM-Hard benchmark as it is hard to derive a generally applicable tool function for all test cases. We mainly experiment LATM with GPT-4 as the tool maker since Cai et al. (2023) found that GPT-3.5 fails in all 5 trials on hard tasks like Tracking Shuffled Objects (5). If the generated tool is not general enough or only suitable for training samples, the tool using phase will fail. We perform experiments using GPT-4 and GPT-3.5-Turbo with a sampling temperature of 0 for all settings except PoT and NLEP on GSM-Hard in (b) which use a temperature of 0.5 to increase the sampling diversity. Since task-general PoT and NLEP lack task-specific programming instruction, they may generate non-executable Python programs. We select some settings and give each instance failed at execution up to three additional trials with temperature=0.4 to diversify the possible outputs. No label leakage is involved in this process as only the success of execution is used as a judgement. We report the results with these extra retries on execution failures in ($\rightarrow$). The highest score among each sub-table (a), (b), (c) and (d) is underlined and the best result for each task is in **bold**.

**CodeLlama-7b-Instruct Results.** To investigate the effect of NLEP on compact large language models, we report the results with CodeLlama-7b-Instruct (Rozière et al., 2023) in Table 9. Following the guidance of the instruction-following models[1], we employ a chat session to include task-specific and task-general prompts as previous turns by interleaving the "user" and "assistant" messages with a system message "Provide answers in Python" at the beginning. Hence, we only treat bug-free Python programs that have the desired results after execution as correct answers, regardless of natural language outputs since we explicitly prompt CodeLlama to generate the answer in Python Unlike the prominent performance of GPT-4, the positive impact of NLEP with CodeLlama-7b-Instruct is diminished due to the much smaller model size and greatly reduced programming capability. Although NLEP prompting outperforms the task-general PoT by a large margin on Chinese Remainder Theorem and Scheduling Meeting benchmarks, a non-negligible performance gap is observed between NLEP and task-specific PoT on most tasks.

To further investigate the benefits of NLEP, we fine-tune a CodeLlama-7b (Rozière et al., 2023) model using NLEP-style instances, resulting in a variant that we term NLEP-CodeLlama. Note that our training corpus does not include specific evaluation tasks. During the evaluation phase, we

---

[1] https://github.com/facebookresearch/codellama

|  | Tracking Shuffled Objects (7) | Dyck Language | Word Sorting | Chinese Remainder Theorem | Scheduling Meeting | GSM-Hard |
|---|---|---|---|---|---|---|
| (a) Task-Specific Prompting: CodeLlama-7b-Instruct | | | | | | |
| PoT | 95.6 | 15.2 | 78.0 | 100.0 | 32.0 | 23.9 |
| (b) Task-General Prompting: CodeLlama-7b-Instruct | | | | | | |
| PoT | 21.2 | 0.8 | 98.0 | 0.0 | 4.0 | 22.9 |
| NLEP (Ours) | 30.0 | 0.8 | 93.2 | 18.8 | 24.8 | 15.2 |
| (c) Zero-shot Prompting: NLEP Trained CodeLlama-7b | | | | | | |
| Zero-shot | 84.4 | 1.2 | 98.4 | 0.0 | 34.4 | 16.8 |

**Table 9:** Performance on six math and symbolic reasoning tasks. We directly prompt CodeLlama-7b-Instruct (Rozière et al., 2023) with task-specific and task-general demonstrations in (a) and (b). The corresponding experimental setup remains consistent with these outlined in Table 2 except we employ a chat session. In this instance, we incorporate the in-context demonstrations as previous turns by interleaving the "user" and "assistant" messages. We further train CodeLlama-7b (Rozière et al., 2023) with NLEP examples and report the zero-shot performance in (c). We adhere to the configuration employed in the GPT-series experiments, wherein we prepend the in-context demonstrations before each test instance.

adopt zero-shot prompting strategy, where the model is provided with only test instances without in-context demonstrations. As presented in Table 9(c), zero-shot NLEP-CodeLlama exhibits consistent performance improvement on 5 of 6 tasks. The only exception is the Chinese Remainder Theorem benchmark, which is notably more complex in nature. Remarkably, zero-shot NLEP-CodeLlama demonstrates superior performance on Word Sorting benchmark when compared to task-general CoT prompted GPT-3.5-Turbo, and outperforms NLEP prompted GPT-3.5-Turbo on Tracking Shuffled Objects (7) benchmark, despite a considerably lower parameter size.

**Game of 24 Results.** We present the detailed experimental results on the Game of 24 benchmark in Table 10. The effect of extra retries described in Section 4 is highlighted with ($\rightarrow$).

| setting | Task-Specific | | | | | | Task-General | |
|---|---|---|---|---|---|---|---|---|
| | IO | CoT | IO (best of 100) | CoT (best of 100) | ToT (b=1) | ToT (b=5) | PoT | NLEP (ours) |
| Game of 24 | 7.3[†] | 4.0[†] | 33.0[†] | 49.0[†] | 45.0[†] | **74.0**[†] | 52 ($\rightarrow$52) | 63 ($\rightarrow$66) |

**Table 10:** Performance on Game of 24 benchmark. [†] stands for results from Yao et al. (2023a).

**Execution Failure Analysis.** We present the execution failure statistics of code-based reasoning strategies in Table 11. The effect of extra retries described in Section 4 is highlighted with ($\rightarrow$). Note that different from task-specific PoT with demonstrations showing how to return the desired outputs in Python program, e.g.,

```python
# Python code, return ans
Alice = " striker "
Bob = " right  winger"
Claire = " left  winger"
Dave = "benchwarmer"
Eve = "goalkeeper"
Fred = " center  midfielder "
Gertrude = " cheerleader "
Eve, Claire = Claire , Eve
Gertrude, Alice = Alice , Gertrude
Fred, Bob = Bob, Fred
Dave, Fred = Fred, Dave
Fred, Bob = Bob, Fred
Bob, Eve = Eve, Bob
Claire , Alice = Alice , Claire
ans = Gertrude
```

we need to design rules to extract the target answers from the execution results of task-general PoT and NLEP since they are allowed to generate free-from outputs. For example, given the generated programs,

```
# Step 1: Import necessary built-in libraries
# No need to import

# Step 2: Define necessary functions that generally solve this type of problem
def swap_positions(positions, swaps):
    for swap in swaps:
        positions[swap[0]], positions[swap[1]] = positions[swap[1]], positions[swap[0]]
    return positions

# Step 3: Define constant variables for the task
positions = {
    "Alice": "striker",
    "Bob": "right winger",
    "Claire": "left winger",
    "Dave": "benchwarmer",
    "Eve": "goalkeeper",
    "Fred": "center midfielder",
    "Gertrude": "cheerleader"
}

swaps = [
    ("Eve", "Claire"),
    ("Gertrude", "Alice"),
    ("Fred", "Bob"),
    ("Dave", "Fred"),
    ("Fred", "Bob"),
    ("Bob", "Eve"),
    ("Claire", "Alice")
]

# Step 4: Print an answer in natural language.
final_positions = swap_positions(positions, swaps)
print(f"At the end of the match, Gertrude is playing {final_positions['Gertrude']}.")
```

we need to extract "striker", the target answer, from the execution results "At the end of the match, Gertrude is playing striker.".

Although task-specific PoT explicitly instructs the model to generate a required line of code `ans={desired_result}` to capture the final result by adding

```
# Python code, return ans
```

in the instruction and explicitly provides task-specific examples in the prompt, e.g.,

```
# Python code, return ans
... ...
ans = Gertrude
```

we found that GPT models sometimes fail to accomplish the requirement. On Tracking Shuffled Objects (7) benchmark, only 17 programs out of 250 test cases generated by GPT-3.5 successfully contain the required line of code, i.e., `ans={desired_result}`, which explains the huge number of failures (233). In addition to already having task-specific programming examples, the inability to distinguish between the erroneous programs and lack of required line of code is another reason why we do not apply the error retries on task-specific PoT.

| | GPT-4 | | | GPT-3.5-Turbo | | |
|---|---|---|---|---|---|---|
| | Task-Specific | Task-General | | Task-Specific | Task-General | |
| | PoT | PoT | NLEP | PoT | PoT | NLEP |
| Track Shuffled Objects (7) | 0 | 0 | 0 | 233 | 26 | 24 |
| Dyck Language | 16 | 24 | 10 | 32 | 81 | 26 |
| Word Sort | 0 | 0 | 0 | 0 | 0 | 0 |
| Chinese Remainder Theorem | 0 | 32 ($\rightarrow$0) | 37 ($\rightarrow$6) | 0 | 46 ($\rightarrow$7) | 4 ($\rightarrow$0) |
| Scheduling Meeting | 0 | 3 ($\rightarrow$0) | 43 ($\rightarrow$0) | 2 | 15 ($\rightarrow$2) | 36 ($\rightarrow$0) |
| GSM-Hard | 17 | 6 | 8 | 31 | 464 ($\rightarrow$145) | 95 ($\rightarrow$13) |

**Table 11:** Execution failure statistics on six math and symbolic reasoning tasks. Results with extra reties are reported in ($\rightarrow$). For task-specific PoT, we report the execution error statistics with None as the return value of `safe_execute()` function following the source code of Chen et al. (2022): `https://github.com/wenhuchen/Program-of-Thoughts/blob/main/tool.py`. It includes instances where the generated programs do not contain the required line of code: `ans={desired_result}`, which are explicitly required in the instruction and given in the prompt demonstration. Under this scenario, we cannot capture the execution results of task-specific PoT.

# C PROMPTS FOR TASK-GENERAL STRATEGIES

## C.1 PROMPTS FOR TABLE 2 AND 4

We list the prompts for the task-general chain-of-thought (CoT), our implementation of program-of-thoughts (PoT) and the proposed natural language embedded programs (NLEP) strategies in the following code segments. They share the same two NLEP examples (one is for natural language reasoning and the other is for mathematical reasoning) but with different forms of intermediate reasoning steps (e.g., code, text etc.) to evaluate the average performance of different strategies.

**Prompt for task-general chain-of-thought (CoT) in Table 2 and 4.** The detailed intermediate natural language reasoning chains are generated by prompting GPT-4 given the input and target output.

```
'''
Answer the problem based on the given instruction and input.

### Instruction: Identify the odd one out.
### Input: Twitter, Instagram, Telegram
### Answer:
Let's think step by step.
1. Start by understanding the task instruction. The task is to identify
the odd one out from a given list.
2. Look at the input. The input consists of three items: Twitter,
Instagram, and Telegram.
3. Identify what these items are. Twitter and Instagram are social media
platforms, while Telegram is a messaging app.
4. Compare the items to find the odd one out. Twitter and Instagram are
primarily used for sharing information, images, and videos. On the other
hand, Telegram is mainly used for instant messaging and voice-over-IP
service.
5. Determine the odd one out based on the comparison. In this case,
Telegram is the odd one out because it serves a different primary
function compared to Twitter and Instagram.
6. Formulate the target output. The target output should clearly state
that Telegram is the odd one out and provide the reason why it is so. The
reason being that Twitter and Instagram are social media platforms mainly
for sharing information, images, and videos while Telegram is a cloud-
based instant messaging and voice-over-IP service.
The correct answer is Telegram.

### Instruction: Use the given data to calculate the median.
### Input: [2, 3, 7, 8, 10]
### Answer:
Let's think step by step.
1. Start by understanding the task, which is to calculate the median of a
given data set. The median is the middle value in a sorted, ascending or
descending, list of numbers.
2. Look at the given input, which is a list of numbers: [2, 3, 7, 8, 10].
3. Notice that the list is already sorted in ascending order. If it wasn'
t, the first step would be to sort it.
4. Understand that to find the median, we need to find the middle value.
If the list has an odd number of observations, the median is the middle
number. If the list has an even number of observations, the median is the
average of the two middle numbers.
5. Count the number of values in the list. There are 5 values, which is
an odd number, so the median will be the middle value.
6. Identify the middle value. Since there are 5 values, the middle value
is the 3rd value when counting from either end.
7. Find the 3rd value in the list, which is 7.
8. Conclude that the median of the given data set is 7.
The correct answer is 7.
'''
```

**Prompt for task-general NLEP in Table 2 and 4.** The intermediate program reasoning chains are generated by prompting GPT-4 given the input and target output.

```
Write a bug-free Python program that can generate the answer to the given instruction when correctly executed.

### Instruction: Identify the odd one out.
### Input: Twitter, Instagram, Telegram
### Python program:
```

```python
# Step 1: Import necessary built-in libraries
from collections import OrderedDict

# Step 2: Define necessary functions that generally solve this type of problem
def find_odd_one_out(services, input_services):
    descriptions = [services[service] for service in input_services]
    for description in descriptions:
        if descriptions.count(description) == 1:
            return input_services[descriptions.index(description)]
    return None

# Step 3: Define constant variables for the task
services = OrderedDict([
    ("Twitter", "a social media platform mainly for sharing information, images and videos"),
    ("Instagram", "a social media platform mainly for sharing information, images and videos"),
    ("Telegram", "a cloud-based instant messaging and voice-over-IP service"),
])

input_services = ["Twitter", "Instagram", "Telegram"]

# Step 4: Print an answer in natural language.
odd_one_out = find_odd_one_out(services, input_services)
if odd_one_out:
    other_services = [service for service in input_services if service != odd_one_out]
    print(f"The odd one out is {odd_one_out}. {other_services[0]} and {other_services[1]} are {services[other_services[0]]}
        while {odd_one_out} is {services[odd_one_out]}.\nThe correct answer is {odd_one_out}.")
```

```
### Instruction: Use the given data to calculate the median.
### Input: [2, 3, 7, 8, 10]
### Python program:
```

```python
# Step 1: Import necessary built-in libraries
# No need to import

# Step 2: Define necessary functions that generally solve this type of problem
def calculate_median(data):
    data.sort()
    length = len(data)
    if length % 2 == 0:
        return (data[length//2] + data[length//2 - 1]) / 2
    else:
        return data[length//2]

# Step 3: Define constant variables for the task
data = [2, 3, 7, 8, 10]

# Step 4: Print an answer in natural language.
median = calculate_median(data)
print(f"To find the median of a data set, we need to arrange the data in ascending order and then find the middle value. In
    this case, the given data is already arranged in ascending order. Since there are {len(data)} values in the data set,
    the median will be the middle value, which is the {len(data)//2 + 1}rd value. Hence, the median of the given data set
    is {median}.\nThe correct answer is {median}.")
```

**Prompt for our task-general program-of-thoughts (PoT) implementation in Table 2 and 4.** The intermediate program reasoning chains are obtained from NLEP. We keep the key implementation steps of Python programs but eliminate the step-by-step programming instructions and detailed natural language elaboration as an ablation.

```
Write a bug-free Python program that can generate the answer to the given instruction when correctly executed.

### Instruction: Identify the odd one out.
### Input: Twitter, Instagram, Telegram
### Python program:
```

```python
softwares = {
    'social media': ['twitter', 'instagram'],
    'communication': ['telegram']
}
for genre, apps in softwares.items():
    if len(apps) == 1:
        print(apps[0])
```

```
### Instruction: Use the given data to calculate the median.
### Input: [2, 3, 7, 8, 10]
### Python program:
```

```
‘‘‘
data = [2, 3, 7, 8, 10]
data.sort()
length = len(data)
if length % 2 == 0:
    print((data[length//2] + data[length//2 − 1]) / 2)
else:
    print(data[length//2])
‘‘‘
```

## C.2 PROMPTS FOR TABLE 3

The Game of 24 task is much more challenging and we replace the first example in Appendix C.1 with a word sorting example to elicit stronger reasoning ability.

**Prompt for task-general NLEP in Table 3.** The intermediate program reasoning chains are generated by prompting GPT-4 given the input and target output.

```
Write a bug−free Python program that can generate the answer to the given instruction when correctly executed.

### Instruction: Arrange the following words to make the longest possible word.
### Input: the, had, not, been
### Python program:
‘‘‘
# Section 1: Define necessary functions and calculate intermediate variables
def longest_word(words):
    from itertools import permutations
    all_words = [''.join(p) for p in permutations(''.join(words))]
    all_words.sort(key=len, reverse=True)
    with open('english_words.txt') as word_file:  # Assuming you have a list of english words
        english_words = set(word.strip().lower() for word in word_file)
    for word in all_words:
        if word.lower() in english_words:
            return word
    return None

# Section 2: Define constant variables
words = ["the", "had", "not", "been"]

# Section 3: Insert variables in text outputs using f−strings.
longest = longest_word(words)
if longest:
    print(f"The longest word that can be made from the letters in the words \"{', '.join(words)}\" is \"{longest}\".")
‘‘‘

### Instruction: Use the given data to calculate the median.
### Input: [2, 3, 7, 8, 10]
### Python program:
‘‘‘
# Step 1: Import necessary built−in libraries
# No need to import

# Step 2: Define necessary functions that generally solve this type of problem
def calculate_median(data):
    data.sort()
    length = len(data)
    if length % 2 == 0:
        return (data[length//2] + data[length//2 − 1]) / 2
    else:
        return data[length//2]

# Step 3: Define constant variables for the task
data = [2, 3, 7, 8, 10]

# Step 4: Print an answer in natural language.
median = calculate_median(data)
print(f"To find the median of a data set, we need to arrange the data in ascending order and then find the middle value. In
    this case, the given data is already arranged in ascending order. Since there are {len(data)} values in the data set,
    the median will be the middle value, which is the {len(data)//2 + 1}rd value. Hence, the median of the given data set
    is {median}.")
‘‘‘
```

**Prompt for our task-general program-of-thoughts (PoT) implementation in Table 3.** The intermediate program reasoning chains are obtained from NLEP. We keep the key implementation steps

of Python programs but eliminate the step-by-step programming instructions and detailed natural language elaboration as an ablation.

```python
Write a bug-free Python program that can generate the answer to the given instruction when correctly executed.

### Instruction: Arrange the following words to make the longest possible word.
### Input: the, had, not, been
### Python program:
```
def longest_word(words):
    from itertools import permutations
    all_words = [''.join(p) for p in permutations(''.join(words))]
    all_words.sort(key=len, reverse=True)
    with open('english_words.txt') as word_file:  # Assuming you have a list of english words
        english_words = set(word.strip().lower() for word in word_file)
    for word in all_words:
        if word.lower() in english_words:
            return word
    return None

words = ["the", "had", "not", "been"]

longest = longest_word(words)
if longest:
    print(longest)
```

### Instruction: Use the given data to calculate the median.
### Input: [2, 3, 7, 8, 10]
### Python program:
```
data = [2, 3, 7, 8, 10]
data.sort()
length = len(data)
if length % 2 == 0:
    print((data[length//2] + data[length//2 - 1]) / 2)
else:
    print(data[length//2])
```
```

## C.3   Prompts for NLEP in Table 5 and Figure 4

For experiments in TruthfulQA and VicunaQA, we added the following example into the NLEP prompts shown in Appendix C.1 to encourage generating more detailed responses:

```python
# Write a bug-free Python program that can generate the answer to the given instruction when correctly executed. Do not ask
    for user input. For reasoning tasks, define functions first and then define variables. For knowledge intensive tasks,
    define variables before defining functions. Do not define any variable that directly stores the final answer. If there
    is a knowledge definition step, use dictionaries to store both the knowledge and detailed explanation.

### Instruction: Discuss the causes of the Great Depression
### Input: None
### Python program:
```
# Step 1: Import necessary built-in libraries
# No need to import

# Step 2: Define dictionaries storing detailed knowledge about the grat depression
depression_name = "The Great Depression"
depression_period = "1929-1939"
depression_countries = "the United States and countries around the world"
depression_causes = {
    "Stock Market Crash of 1929": "In October of 1929, the stock market experienced a significant fall that wiped out
        millions of investors. This event is considered by many to be the initial trigger of the Great Depression.",
    "Overproduction": "During the 1920s, many industries produced more goods than consumers wanted or could afford. This
        ultimately led to a decline in demand for goods, causing job loss, lower wages, and business failure.",
    "High Tariffs and War Debts": "Protectionist trade policies in the form of high tariffs led to a decline in global trade,
        as other countries retaliated with tariffs of their own. Additionally, many countries were struggling to repay war
        debts, which led to economic instability.",
    "Bank Failures": "As demand for goods declined, many banks began to fail, causing a loss of confidence in the banking
        system. This led to a massive withdrawal of money from banks, causing even more banks to fail.",
    "Drought Conditions": "The Dust Bowl was a severe drought and dust storm that hit the Great Plains region of the United
        States in the 1930s. This had a significant impact on agriculture, causing many farmers to lose their land and
        livelihoods which worsened the effects of the depression."
}
```
```

```python
# Step 3: Define necessary functions that generally solve this type of problem
# Do not need to define functions

# Step 4: Print the answer and explain in natural language by calling the information in the defined knowledge dictionary 'depression_causes'
print(f"{depression_name} was a period of economic decline that lasted from {depression_period}, making it the longest-lasting depression in modern history. It affected not only {depression_countries}, causing substantial social and economic upheaval.\n")
print(f"There were several major causes of {depression_name}, which include:\n")

# List causes and explanations in 'depression_causes' with a for-loop.
for i, (cause, description) in enumerate(depression_causes.items(), 1):
    print(f"{i}. {cause} - {description}\n")
print(f"Overall, {depression_name} was caused by a combination of factors, including economic, environmental, and political factors. Its impact was widespread, affecting millions of people around the world.")
```

### Instruction: Identify the odd one out.
### Input: Twitter, Instagram, Telegram
### Python program:
```python
# Step 1: Import necessary built-in libraries
from collections import OrderedDict

# Step 2: Define dictionaries storing detailed knowledge about the main function of each application
services = {
    "Twitter": "a social media platform mainly for sharing information, images and videos",
    "Instagram": "a social media platform mainly for sharing information, images and videos",
    "Telegram": "a cloud-based instant messaging and voice-over-IP service",
}

# Step 3: Define a function that finds the different application
def find_odd_one_out(services, input_services):
    descriptions = [services[service] for service in input_services]
    for description in descriptions:
        if descriptions.count(description) == 1:
            return input_services[descriptions.index(description)]
    return None

# Step 4: Print the answer in natural language by calling the information stored in 'services' and the defined function 'find_odd_one_out'
input_services = ["Twitter", "Instagram", "Telegram"]
odd_one_out = find_odd_one_out(services, input_services)
if odd_one_out:
    other_services = [service for service in input_services if service != odd_one_out]
    print(f"The odd one out is {odd_one_out}. {other_services[0]} and {other_services[1]} are {services[other_services[0]]} while {odd_one_out} is {services[odd_one_out]}.")
```

### Instruction: Calculate the total surface area of a cube with a side length of 5 cm.
### Input: None
### Python program:
```python
# Step 1: Import necessary built-in libraries
# No need to import

# Step 2: Define a function that calculate the surface area of cubes
def calculate_surface_area(side_length):
    return 6 * (side_length ** 2)

# Step 3: Define dictionaries storing the cube information
cube = {
    "side_length": 5  # Side length of the cube
}

# Step 4: Print a step-by-step calculation answer in natural language using the defined function and varible
side_length = cube["side_length"]
surface_area = calculate_surface_area(side_length)
print(f"The surface area of a cube is found by calculating the area of one of its faces and multiplying it by six (since a cube has six faces). The area of a cube face is simply its side length squared.\n")
print(f"Thus for this particular cube:")
print(f"Surface Area = 6 x (Side Length)\^2")
print(f"             = 6 x ({side_length} cm)\^2")
print(f"             = 6 x {side_length**2} cm\^2")
print(f"             = {surface_area} cm\n")
print(f"The total surface area of this cube is {surface_area} square centimeters.")
```
```

## C.4 Prompts for Table 6 and 7

We use the following prompt for the entailment-based NLEP results in Table 6. The model-free result uses the NLEP prompt shown in C.1.

```python
"""Write a Python function that constructs a decision tree according to the given examples that can generate the correct
    label of the given classification task."""

### Available functions (shared for all tasks):

# Returns whether the hypothesis in entailed by the premise.
def entailment(hypothesis, premise, model, tokenizer):
    proposition = f'{hypothesis} is entailed by {premise}.'
    inputs = tokenizer(proposition, return_tensors="pt", truncation=True, padding=True, max_length=128)
    outputs = model(**inputs)['logits'][0]
    ent_label = int(outputs[0] > outputs[2])
    if ent_label == 1:
        return 'yes'
    else:
        return 'no'

# Use the constructed decision tree to predict the label of the sentence.
def tree_predict(sentence, criterions, tree, model, tokenizer):
    node = tree['root']
    while node not in POSSIBLE_CLASSES:
        ent_label = entailment(criterions[node], sentence, model, tokenizer)
        node = tree[node][ent_label]
    return node

### Task: Movie review classification
### Possible classes: [positive, negative]
### Examples:
"""
- contains no wit, only labored gags
    - [The movie is wise|The movie is not wise|1], [the story is fun|the story is not boring|1], [the review is positive|the
        review is negative|1]
- that loves its characters and communicates something rather beautiful about human nature
    - [The characters are lovely|The characters are awful|0], [the script is touching|the script is dry|0], [the review is
        positive|the review is negative|0]
- on the worst revenge−of−the−nerds cliches the filmmakers could dredge up
    - [The movie is novel|The movie is mostly platitudes|1], [the review is negative|1]
- are more deeply thought through than in most right−thinking films
    - [The takeaway of the movie is profound|The idea of the movie is shallow|0], [the review is positive|the review is
        negative|0]
"""

### Define possible classes
POSSIBLE_CLASSES = ['positive', 'negative']

### Python program:
'''
def get_decision_tree(sentence, model, tokenizer):
    # Step 1: define criterions of the decision tree.
    criterions = [
        'This movie is interesting',
        'The movie has a good script',
        'The characters are awsome',
        'This movie is wise'
    ]

    # Step 2: define the Decision Tree for classification
    tree = {
        'root': 0,
        0: {'yes': 1, 'no': 3},
        1: {'yes': 'positive', 'no': 2},
        2: {'yes': 'positive', 'no': 'negative'},
        3: {'yes': 'positive', 'no': 'negative'}
    }

    return criterions, tree
'''
```

When we test the SST-2 performance based on a generated Cola decision tree in Table 7, we use the following prompt:

```
Write a Python function that constructs a decision tree according to the given examples that can generate the correct label
    of the given classification task.
```

```python
### Available APIs(shared for all tasks):

# Returns whether the hypothesis in entailed by the premise.
def entailment(hypothesis, premise, model, tokenizer):
    proposition = f'{hypothesis} is entailed by {premise}.'
    inputs = tokenizer(proposition, return_tensors="pt", truncation=True, padding=True, max_length=128)
    outputs = model(**inputs)['logits'][0]
    ent_label = int(outputs[0] > outputs[2])
    if ent_label == 1:
        return 'yes'
    else:
        return 'no'

# Use the constructed decision tree to predict the label of the sentence.
def tree_predict(sentence, criterions, tree, model, tokenizer):
    node = tree['root']
    while node not in POSSIBLE_CLASSES:
        ent_label = entailment(criterions[node], sentence, model, tokenizer)
        node = tree[node][ent_label]
    return node

### Task: Grammar correctness classification
### Possible classes: ['accpetable', 'unacceptable']

### Define possible classes
POSSIBLE_CLASSES = ['accpetable', 'unacceptable']

### Decision Tree Logic:
- If verbs are not correctly constructed, the sentence is immediately labeled as unacceptable.
- If verbs are correct:
    The tree then checks if the sentence has correct punctuation
    - If incorrect, label the sentence as unacceptable
    - If correct:
        The next criterion to be assessed is the subject-verb agreement.
        - If subject and verb disagree, label the sentence as unacceptable.
        - If they agree, check for sentence fragments.
            - If the sentence is a fragment, label it as unacceptable.
            - If it is not a sentence fragment, label the sentence as acceptable.

### Python code for the decision tree:

``` python
def get_decision_tree(sentence, model, tokenizer):
    # Step 1: define criterions of the decision tree
    criterions = {
        'correct_verbs': 'The verbs are correctly constructed in the sentence',
        'correct_punctuation': 'The sentence is punctuated correctly',
        'subject_verb_agreement': 'The subject and verb agree in the sentence',
        'no_sentence_fragments': 'The sentence is not a fragment',
    }

    # Step 2: define the balanced decision tree for this classification task
    tree = {
        'root': 'correct_verbs',
        'correct_verbs': {'yes': 'correct_punctuation', 'no': 'unacceptable'},
        'correct_punctuation': {'yes': 'subject_verb_agreement', 'no': 'unacceptable'},
        'subject_verb_agreement': {'yes': 'no_sentence_fragments', 'no': 'unacceptable'},
        'no_sentence_fragments': {'yes': 'acceptable', 'no': 'unacceptable'}
    }

    return criterions, tree
```
```

The input format of target tasks is

```
### Task: Grammar correctness classification
### Possible classes: [acceptable, unacceptable]
```

# D  IMPLEMENTATION DETAILS FOR TASK-SPECIFIC STRATEGIES

We detail the few-shot chain-of-thought (CoT) and program-of-thought (PoT) prompting under the task-specific setting in Tables 2 and 4:

- **Tracking Shuffled Objects (7).** We use the same 3-shot examples as used by previous work for both task-specific CoT and PoT. The three examples are related to Tracking Shuffled Objects (3) and the models need to learn from demonstrations and generalize to seven objects test cases. The difference between CoT and PoT lies on the format of intermediate reasoning: CoT adopts natural language as the reasoning chains while we transform the thought process into concise Python code for PoT.

- **Dyck Language.** We cite the results of CoT from LATM (Cai et al., 2023) and transform the reasoning steps of the 3-shot examples used by previous chain-of-thought work into Python code for PoT. In order to evaluate the generalization ability of program-of-thought prompting, we try to avoid directly giving generally applicable Python program that can be used for all test instances.

- **Word Sorting.** We cite the results of CoT from LATM (Cai et al., 2023) and transform the reasoning steps of the 3-shot examples used by previous chain-of-thought work into Python code for PoT. Since the task can be effectively resolved by just few lines of code, i.e., read in the given input and use `sorted()` function, e.g.,

```python
# Python code, return ans
words = ['oven', 'costume', ' counterpart ']
ans = " ". join ( sorted (words))
```

  it can be regarded that the generally applicable tool is already given in the input prompt.

- **Chinese Remainder Theorem.** We cite the results of CoT from LATM (Cai et al., 2023). We build the in-context examples (3-shot) with the first three successful instances of task-general PoT as we construct the Python reasoning chains from the generated programs of task-general PoT with GPT-4. Indeed, for this complicated task, the provided program in the demonstration can also be regarded as a generally applicable tool. That's a main reason why task-specific PoT can obtain 100% accuracy on this benchmark.

- **Scheduling Meeting.** We cite the results of CoT from LATM (Cai et al., 2023). We build the in-context examples (3-shot) with the first three successful instances of task-general PoT as we construct the Python reasoning chains from the generated programs of task-general PoT with GPT-4. However, unlike giving the "ground-truth" problem solving tool for Chinese Remainder Theorem, the provided Python reasoning chains can only derive the correct answer for each specific demonstration question but can not be directly applied to all scenarios due to the complexity of the problem. We hope to compare this setup with Chinese Remainder Theorem and evaluate the performance of task-specific PoT on complicated tasks through different in-context learning demonstrations.

- **GSM-Hard.** We use the same 8-shot examples as used by previous work on GSM8K dataset for CoT GSM-Hard. For PoT, we adopt the 9-shot examples on GSM8K dataset from program-of-thought (Chen et al., 2022) containing concise Python code as reasoning chains.

- **StrategyQA.** We remove 1 example that appears in the development set from the 6-shot demonstration of previous work (Lyu et al., 2023) for CoT. As PoT is not designed and applied for natural language question answering task, we did not reproduce task-specific PoT results on StrategyQA benchmark.

# E   EVALUATION PROMPTS FOR VICUNAQA

We have two metrics for VicunaQA. The first metric assesses the level of details and biases to long responses generated by GPT-4, while the other metric does not ask for details.

## E.1   EVALUATION PROMPT ASKING FOR DETAILS.

prompt = f ''' [Question]\n{ques_str}

[The Start of Assistant 1's Answer]\n{gpt4_res}
\n[The End of Assistant 1's Answer]

[The Start of Assistant 2's Answer]\n{target_res}
\n[The End of Assistant 2's Answer]

[System]
We would like to request your feedback on the performance of two AI assistants in response to the user question displayed above.\nPlease rate the helpfulness, relevance, accuracy, level of details of their responses. Each assistant receives an overall score on a scale of 1 to 10, where a higher score indicates better overall performance.\nPlease first output a single line containing only two values indicating the scores for Assistant 1 and 2, respectively. The two scores are separated by a space. In the subsequent line, please provide a comprehensive explanation of your evaluation, avoiding any potential bias and ensuring that the order in which the responses were presented does not affect your judgment. '''

## E.2   EVALUATION PROMPT NOT ASKING FOR DETAILS.

prompt = f ''' [Question]\n{ques_str}

[The Start of Assistant 1's Answer]\n{gpt4_res}
\n[The End of Assistant 1's Answer]

[The Start of Assistant 2's Answer]\n{target_res}
\n[The End of Assistant 2's Answer]

[System]
We would like to request your feedback on the performance of two AI assistants in response to the user question displayed above.\nPlease rate the relevance and accuracy of their responses. Each assistant receives an overall score on a scale of 1 to 10, where a higher score indicates better overall performance.\nPlease first output a single line containing only two values indicating the scores for Assistant 1 and 2, respectively. The two scores are separated by a space. In the subsequent line, please provide a comprehensive explanation of your evaluation, avoiding any potential bias and ensuring that the order in which the responses were presented does not affect your judgment. Do not bias on either longer or shorter answers. '''

## E.3   CALCULATION OF LENGTH BIAS

Suppose we have $N$ evaluation cases, each receiving 2 candidate responses. A GPT-4 scorer decides the winner between the candidates. $a$ stands for the number of cases where a candidate response with more tokens wins. The length bias is calculated by

$$lb = 1 - |\frac{a}{N} - 0.5| * 2 \tag{1}$$

