# OpenReview forum: "Natural Language Embedded Programs for Hybrid Language Symbolic Reasoning"
_ICLR.cc/2024/Conference — ICLR 2024 Conference Withdrawn Submission_

### Official Review · Reviewer_ZKfU · 2023-10-24

**Soundness:** 3 good
**Presentation:** 2 fair
**Contribution:** 2 fair
**Rating:** 3
**Confidence:** 3

**Summary:**

The paper presented a prompting strategy to LLMs to generate interpretable programs to symbolic/math/text reasoning tasks.

**Strengths:**

The author thoroughly reviewed and combined existing experience in prompting the LLM for better reasoning results, e.g. CoT, PoT.

**Weaknesses:**

While the approach is well presented, the overall contribution is less clear to the reviewer.

* The author claimed that NLEP is a unified framework for math/symbolic reasoning, NLU, and instruction following tasks. Since the entire model, except the text classification, is just prompting the LLM, it's unified naturally. On the other hand, from Table 1, each domain may share the same prompt, while the prompts still need to be different cross-domain. It's super reasonable for sure, but the unified framework is further weakened.

* Fundamentally, the experiment section compares the prompting strategies. It's better to compare the content in the prompts side by side.

The reviewer suggests improving the presentation of the method and emphasizing the difference to the existing models with concise and concrete facts. It's possible that the reviewer misunderstand the contribution of this work due to the presentation quality.

**Questions:**

1. Which part of the experiment evaluates the NLEP's performance on the instruction following?

2. What do numbers in parenthesis mean in Table 6 header?

3. What is the foundation model for multi-class prompting in Table 6?

4. it is quite unclear that whether the NLEP is a prompting strategy or a refined decoding space. In Figure 1, it seems the only input to the LLM is a vanilla question. In appendix C, the contents are titled "prompt for task-general CoT/NLEP", but then described as generated results by LLM. In addition, if these results are referred to as "intermediate program", a description of full process should be available. Is the program generated then executed on a python interpreter for the NLEP and PoT while the CoT just directly predict the results? If so, it's an important difference that should be emphasized.

---

### Official Review · Reviewer_2MWH · 2023-10-30

**Soundness:** 2 fair
**Presentation:** 3 good
**Contribution:** 1 poor
**Rating:** 5
**Confidence:** 3

**Summary:**

The paper proposes using natural language embedded programs (NLEPs) as a unified framework for combining language-based reasoning with symbolic computations to solve tasks requiring both capabilities. NLEPs are generated by prompting large language models (LLMs) with general demonstrations as prompts. The generated NLEP code is then executed and the output is captured as the response.
Tasks Evaluated including Math and symbolic reasoning, Question answering and instruction following, Text classification, shows the improvement over strong baselines across the diverse tasks.

**Strengths:**

1. The proposed natural language embedded programs (NLEP) framework integrates the strengths of both natural language reasoning and symbolic computation. It bridges the gap between symbolic and linguistic representations but also offers a more comprehensive approach to problem-solving.
2. One of the standout features of the NLEP approach is its adaptability. With just a task-general prompt, the framework can tackle a diverse set of tasks, from math and symbolic reasoning to text classification and question answering. This adaptability suggests that the framework is not overly specialized and can be applied to a wide range of NLU challenges.
3. In an era where the interpretability of machine learning models is of paramount importance, the NLEP framework shines by generating Python programs that are not only human-readable but also allow for a deeper understanding of the model's reasoning process. This transparency is crucial for applications where understanding the decision-making process is as important as the decision itself. Beyond just interpretability, the generated programs offer a tangible advantage in terms of verification. By being able to inspect and execute the generated code, researchers and practitioners can verify the intermediate steps of reasoning. This is a significant step forward in ensuring the reliability and trustworthiness of AI-driven solutions.
4. The paper demonstrates that the NLEP approach can outperform strong baselines across various tasks. This not only validates the efficacy of the proposed method but also reveals its potential as a leading approach in the domain of natural language understanding.

**Weaknesses:**

My main concern lies in the incremental advancement Over Prior Work. While the NLEP framework undeniably brings advancements in enhancing the reasoning capabilities of large language models, its novelty in relation to prior work raises questions. The authors acknowledge in the paper regarding to the similarities with existing methods such as the program-of-thought (PoT; Chen et al., 2022) and program aided language models (PAL; Gao et al., 2023). Although NLEPs introduce additional programming elements like packages, data types/structures, and functions, the core contribution appears to be more of an engineering-style extension rather than a groundbreaking conceptual leap. It would be beneficial to see a more in-depth discussion on the unique contributions of NLEP beyond these incremental enhancements.

**Questions:**

N/A

---

### Official Review · Reviewer_pxPW · 2023-11-01

**Soundness:** 2 fair
**Presentation:** 3 good
**Contribution:** 2 fair
**Rating:** 3
**Confidence:** 3

**Summary:**

This work proposes natural language embedded programs (NLEP) as a unified framework to perform math/symbolic reasoning. This approach prompts a language model in a four-step form, tool-using, knowledge extraction, function definition, and answer printing, to generate a runnable Python program. This method can improve upon strong baselines over a range of tasks.

**Strengths:**

Originality: 2/5

There are a few related works in the neural symbolic works of literature that instead of using the GPT-4 model to generate Python code with prompting, extract necessary facts and perform reasoning over these facts.

Quality: 3.5/5

This work has been sufficiently evaluated on a few benchmarks with three baselines.

Clarity: 2.5/5

The presentation of the paper is somewhat hard to follow. A flowchart diagram would be nice to represent which part of the framework is up to changes and which part is not.

Significance: 3.5/5
Combining neural models with symbolic programs is an important task to solve realistic problems which requires both perception and reasoning capabilities.

**Weaknesses:**

In the related work, the neural symbolic works are not discussed and compared sufficiently. For example, Scallop [1] is a language for combining perceptual and reasoning capabilities; Logic-LM [2] also utilizes a logic solver for LLM-extracted knowledge.

[1] Li, Ziyang, Jiani Huang, and Mayur Naik. "Scallop: A Language for Neurosymbolic Programming." Proceedings of the ACM on Programming Languages 7.PLDI (2023): 1463-1487.

[2] Pan, Liangming, et al. "Logic-lm: Empowering large language models with symbolic solvers for faithful logical reasoning." arXiv preprint arXiv:2305.12295 (2023).

**Questions:**

What is the advantage and disadvantages comparing your work against the neural symbolic approaches?

---

### Official Review · Reviewer_T68U · 2023-11-04

**Soundness:** 3 good
**Presentation:** 3 good
**Contribution:** 3 good
**Rating:** 8
**Confidence:** 4

**Summary:**

This paper proposed natural language embedded programs (NLEP), which is a general method for several natural language reasoning and understanding tasks, such as math reasoning, question answering and text classification. NLEP first uses natural language comments as a step-by-step guide for program generation, by using programs as solutions, it prints out a natural language response simply as a string. Experiments are conducted on 16 language understanding tasks, and results show that NLEP is comparable with CoT and PoT methods using task-specific prompts, and outperforms such methods when only task-general prompts are considered.

**Strengths:**

S1: Despite being simple, the proposed method is quite interesting and novel. It is also a good complement to the tool use of LLMs, which has been a popular domain lately;
S2: The evaluation is very comprehensive, 16 tasks from three domains are evaluated in this work, and the proposed method has been shown to be comparable to or better than previous methods (e.g., CoT, ToT) on a majority of the tasks;
S3: One of the key contributions of NLEP is task-general prompt, which could be very helpful in making LLMs less dependent or sensitive to specific prompts.

**Weaknesses:**

W1: Some ablation study for NLEP would have been nice to understand the key components that make it work. For example, a study on how important are those step-by-step natural language comments;
W2: It seems to me that such methods would probably be hard to extend to natural language generation tasks (e.g., summarization, storytelling, etc), but this is not to discredit this work as language reasoning itself is already an interesting and important domain.

In addition, it would be great if the authors can answer the questions in the "questions" section.

**Questions:**

Q1. For structured knowledge shown in Figure 1, did the model generate them or it is automatically generated via some external code (e.g., loading from a json file)?
Q2. Is it possible to also use task-specific prompts to further improve the performance of NLEP and surpass CoT, PoT, and ToT methods completely?
Q3. For the classification tasks, I like the idea of using decision trees to make things discrete and interpretable. Are there any other ways of using NLEP for classification besides decision tree?